# The Effect of Anticoagulants, Temperature, and Time on the Human Plasma Metabolome and Lipidome from Healthy Donors as Determined by Liquid Chromatography-Mass Spectrometry

**DOI:** 10.3390/biom9050200

**Published:** 2019-05-23

**Authors:** Manoj Khadka, Andrei Todor, Kristal M. Maner-Smith, Jennifer K. Colucci, ViLinh Tran, David A. Gaul, Evan J. Anderson, Muktha S. Natrajan, Nadine Rouphael, Mark J. Mulligan, Circe E. McDonald, Mehul Suthar, Shuzhao Li, Eric A. Ortlund

**Affiliations:** 1Emory Integrated Lipidomics Core, Emory University School of Medicine, Atlanta, GA 30322, USA; manoj.khadka@emory.edu (M.K.); kristal.m.maner-smith@emory.edu (K.M.M.-S.); jennifer.k.colucci@emory.edu (J.K.C.); 2Department of Biochemistry, Emory University School of Medicine, Atlanta, GA 30322, USA; 3Department of Medicine, Emory University School of Medicine, Atlanta, GA 30322, USA; andrei.todor@emory.edu (A.T.); vilinh.tran@emory.edu (V.T.); evanderson@emory.edu (E.J.A.); muktha.natrajan@emory.edu (M.S.N.); nroupha@emory.edu (N.R.); mark.mulligan@nyulangone.org (M.J.M.); sli49@emory.edu (S.L.); 4School of Chemistry and Biochemistry, Georgia Institute of Technology, Atlanta, GA 30332, USA; david.gaul@chemistry.gatech.edu; 5Department of Paediatrics, Emory University School of Medicine, Atlanta, GA 30322, USA; circemcd@gmail.com (C.E.M.); mehul.s.suthar@emory.edu (M.S.)

**Keywords:** lipidomics, metabolomics, anticoagulants, vaccine, storage conditions, sample collection

## Abstract

Liquid-chromatography mass spectrometry is commonly used to identify and quantify metabolites from biological samples to gain insight into human physiology and pathology. Metabolites and their abundance in biological samples are labile and sensitive to variations in collection conditions, handling and processing. Variations in sample handling could influence metabolite levels in ways not related to biology, ultimately leading to the misinterpretation of results. For example, anticoagulants and preservatives modulate enzyme activity and metabolite oxidization. Temperature may alter both enzymatic and non-enzymatic chemistry. The potential for variation induced by collection conditions is particularly important when samples are collected in remote locations without immediate access to specimen processing. Data are needed regarding the variation introduced by clinical sample collection processes to avoid introducing artifact biases. In this study, we used metabolomics and lipidomics approaches paired with univariate and multivariate statistical analyses to assess the effects of anticoagulant, temperature, and time on healthy human plasma samples collected to provide guidelines on sample collection, handling, and processing for vaccinology. Principal component analyses demonstrated clustering by sample collection procedure and that anticoagulant type had the greatest effect on sample metabolite variation. Lipids such as glycerophospholipids, acylcarnitines, sphingolipids, diacylglycerols, triacylglycerols, and cholesteryl esters are significantly affected by anticoagulant type as are amino acids such as aspartate, histidine, and glutamine. Most plasma metabolites and lipids were unaffected by storage time and temperature. Based on this study, we recommend samples be collected using a single anticoagulant (preferably EDTA) with sample processing at <24 h at 4 °C.

## 1. Introduction

Recent advancements in sensitive analytical technologies such as liquid-chromatography mass spectrometry (LC-MS) have enabled the identification and quantification of polar and non-polar metabolites in biological samples. Lipidomics approaches entail measuring and assessing the biological roles of the entire spectrum of lipid molecular species in a cell/tissue which can be influenced by expression of proteins involved in lipid metabolism and function, including gene regulation” [1]. Similarly, metabolomics refers to measuring the complete set of metabolites that change according to the physiological, developmental, or pathological state of the cell, tissue, organ, or organism [2]. Metabolomics and lipidomics approaches employ high-resolution mass spectrometry to profile thousands of chemical components in an array of biological samples, making it one of the most powerful technologies to investigate metabolic perturbations in normal biology, disease or other exogenous or endogenous factors. Recently, the Human Metabolome Database (http://www.hmdb.ca/) reported the classification of more than a hundred thousand metabolites [3] which was made possible due to advancement in analytical platforms and computational tools.

The hundreds of thousands of metabolites and lipids [3] that are synthesized via several hundred biochemical reactions drive biochemical processes such as signaling, membrane integrity, inflammation, homeostasis, and many other regulatory functions [4,5,6,7,8,9,10,11]. Cellular metabolic activities can be studied by quantifying changes in lipids at the class, subclass, and molecular species level and discerning the differential metabolites between normal and abnormal cells [12]. Several infectious and non-infectious diseases involve perturbation of biochemical pathways that are reflected in metabolite and lipid levels in tissues, blood, and other biological entities. Thus, change in the normal concentrations of these metabolites and lipids often relates to disease and differential physiological states [13,14,15].

Metabolites can be studied by two major approaches—untargeted and targeted—through utilization of different analytical platforms. Untargeted analyses attempt to broadly profile as many metabolites as possible, while targeted analyses specifically characterize metabolites based on their chemical properties [16]. LC-MS and gas chromatography-mass spectrometry (GC-MS) are most commonly used for comprehensively studying metabolites and lipids in both targeted and untargeted analyses. Although frequently used, the sensitivity and resolution of NMR are inferior to MS and only able to detect highly abundant metabolites [16]. However, MS platforms capture broad metabolite classes with high sensitivity, making them an ideal platform for the comprehensive study of metabolites.

To gain insight into organismal- or organ-level metabolism and physiology, biological fluids such as plasma, serum, urine, cerebrospinal fluid and tissues are often utilized for metabolomics and lipidomics studies. Maintaining the integrity of the metabolite pool is critical to link metabolite identities and concentrations with biology. The collection and processing of clinical samples require a series of physical and chemical processing steps that expose metabolites to degradation and alteration (e.g., enzymatic or oxidation). Processing time, storage condition, shipping time, anticoagulants, and freeze-thaw cycles, among other variables, could introduce analytical and experimental biases that might introduce significant technical challenges to the rigor of clinical research. It can be challenging, even with comprehensive statistical approaches, to differentiate between systematic bias in sample processing and handling vs. experimental group distinction [17]. Confounding factors introduced during sample collection, handling, and processing must be characterized before initiation of lipidomics or metabolomics measurements to prevent the misinterpretation of results and to enhance accuracy [18].

Several studies have been conducted to study the effect of anticoagulants, time, and temperature on metabolites and lipids in biological specimens. Such studies are performed to develop an optimal procedure for clinical sample collection. For example, the extent of the anticoagulant effect on metabolites depends upon metabolite class, anticoagulant concentration, and anticoagulants type. Anticoagulants have the ability to enhance metabolite ionization, increase matrix effects [19,20,21], or introduce chemical noise interfering with metabolites quantitation [22]. Given these complexities, it is not surprising that several studies have reported incoherent findings on the effect of additives, temperatures and time on biological samples [23,24,25,26,27,28]. Some metabolites are resistant to degradation over time, depending on storage temperature. Other metabolites are labile and susceptible to degradation even when stored at conventional processing temperatures (4 °C) [29]. The evaluation of possible biases introduced during sample collection, processing, and handling can further guide in experimental design and reduce the effect of the confounding factors on results. Such studies also assist in the development of standard operating procedures (SOP) for the high throughput omics-based studies. Thus, such studies are critical to assess the effect of sample handling artifacts on biological specimens before conducting large-scale studies.

Here we detail a study to assess the effect of storage time, temperature, and anticoagulants (collection tube types) on healthy donors’ plasma samples in preparation for subsequent vaccine studies conducted through the NIH-funded Vaccine Treatment and Evaluation Unit (VTEU). We also investigate some of the possible physical conditions that samples may encounter during collection, storage, shipping, and handling from collection sites. The findings from this study will be utilized to guide sample collection and processing workflow not only for metabolomics and lipidomics but also at the systems-level including proteomics and transcriptomics (reported elsewhere).

## 2. Materials and Methods

### 2.1. Reagents and Chemicals

All lipid standards were purchased from Avanti Polar Lipids Inc. (Alabaster, AL, USA). HPLC-grade extraction and LC-MS solvents were purchased from Fisher Scientific (Hampton, NH, USA) and Sigma Aldrich (St. Louis, MO, USA).

### 2.2. Lipid Extraction and LC-MS

A total of seventy plasma samples collected from five healthy donors were obtained from an Emory IRB approved healthy donor protocol and identified as D11, D12, D13, D14, and D15. Plasma samples were collected in two different types of tubes containing di-potassium ethylenediaminetetraacetic acid (K_2_EDTA) or a sodium citrate (Na-citrate)-containing cell preparation tube (BD Vacutainer Systems, NJ, USA), respectively. Donor plasma in each tube type was stored for five different lengths of time (0 h (0H), 2 h (2H), 4 h (4H), 8 h (8H), and 24 h (24H)) prior processing. The K_2_EDTA-containing plasma samples were stored at 4 °C or room temperature (22 °C) and the sodium-citrate-containing plasma samples were stored at room temperature (Refer to Figure 1 for detail analytical strategy for metabolomics and lipidomics data). For brevity, the K_2_EDTA samples at 4 °C, K_2_EDTA samples at room temperature and sodium-citrate samples at room temperature will be represented as EDTA_4C, EDTA, and CPT respectively.

Lipids were extracted from the seventy healthy donor plasma samples using the Bligh-Dyer method [30], with slight modifications according to the following brief protocol. A 2:1 *v*/*v* methanol-chloroform mixture (1.8 mL) was added to 200 µL of human plasma and vortexed for 30 min. An additional 1.2 mL methanol: chloroform mixture (1:1 *v*/*v*) was added to the mixture and vortexed briefly. The organic and aqueous phases were separated by adding 0.5 M sodium chloride solution and centrifuged at 2500 rpm for 10 min. The organic fraction was collected in a clean glass vial and was dried and stored at −80 °C until further processing.

Frozen extracts were reconstituted in 100 µL of a 1:1 *v*/*v* methanol: chloroform mixture and 100 µL of internal standard with gentle shaking and brief vortexing. Solvent-only blank solutions were prepared using the reconstituted solvent mixture (methanol: chloroform 1:1 *v*/*v*). Test samples were prepared by diluting LipidoMix 1:100 (Avanti Polar Lipids Inc. Alabaster, AL, USA) in 1:1 *v*/*v* methanol:chloroform to observe baseline lipid separation and the response of the instrument before running the samples. Chromatographic separation was performed using an UltiMate™ 3000 Rapid Separation Binary System (Thermo Scientific; Waltham, MA) and a Waters ACQUITY UPLC BEH C18 column (2.1 × 50 mm, 1.7 µm particle size)). Lipids classes were separated and eluted using binary solvent systems comprised of water: acetonitrile (40:60 *v*/*v*; mobile phase A) and isopropanol: acetonitrile (90:10 *v*/*v*; mobile phase B,) with both phases A and B containing 10 mM ammonium formate and 0.1% formic acid. The gradient flow parameters of mobile phase B were as follows: 0–1 min 40%–45%, 1.0–1.1 min 45%–50%, 1.1–5.0 min 50%–55%, 5.0–5.1 min 55%–70%, 5.1–8.0 min 70%–99%, 8.0–8.1 min 99%–40%, 8.1–9.5 min 40%. The solvent flow was maintained at 0.4 mL/min throughout the LC run. The column temperature was set at 50 °C. Five microliters of sample, from the 200 μL reconstituted volume, was injected into LC-MS for data acquisition in data-dependent acquisition mode (DDA) using a Q Exactive HF mass spectrometer (Thermo Fisher; Waltham, MA) equipped with an electrospray ionization (ESI) source, in both positive and negative ion modes. Full and data-dependent acquisition (DDA) scans data were collected at 120,000 and 15,000 resolution, respectively. The mass spectrometer parameters were as follows: AGC target—1.0E5; maximum IT—250 ms; scan range—*m*/*z* 100 to 1500 for full scan; AGC target—1.0E5; maximum IT—30 ms; TopN—10; isolation window—*m*/*z* 0.4, scan range—*m*/*z* 200 to 2000, NCE—10, 30, 50; Underfill ratio—1.0%, charge exclusion—3-8, > 8; dynamic exclusion—15.0 s for DDA scan modes. The instrument was tuned to a mass accuracy of below ±3 ppm and separation quality was inspected using lipid extract from the commercially available normal human plasma and Lipidomix (Avanti Polar Lipids, Alabaster, AL) before starting the data acquisition for the experimental samples. The chromatogram (Appendix A) shows the lipids separation in positive and negative ionization modes.

### 2.3. Metabolite Extraction and LC-MS

Approximately 100 μL of 2:1 *v*/*v* acetonitrile-water mixture containing internal isotopic mixture was added to the 50 μL of thawed plasma sample, as previously described [31,32]. Following mixing and incubation at 4 °C for 30 min, precipitated proteins were pelleted via centrifugation for 10 min at the 13,500× *g* on a microcentrifuge at 4 °C. Samples were analyzed in triplicate by LC-MS (ThermoScientific High Field (HF) Q Exactive coupled to a Dionex Ultimate 3000 LC system; mass-to-charge ratio (*m*/*z*) range from 85 to 1250 Daltons at 120 K resolution) with a 10 μL injection volume via electrospray ionization. Analyte separation for HILIC was accomplished by a 2.1 mm × 50 mm × 2.5 μm Waters XBridge BEH Amide XP HILIC and an eluent gradient (A = 2% formic acid, B = water, C = acetonitrile) consisting of an initial 1.5 min period of 2.5% A, 22.5% B, 75% C followed by a linear increase to 2.5% A, 77.5% B, 20% C at 4 min and a final hold of 1 min. RPC separation was by 2.1 mm × 50 mm × 3 μm endcapped C_18_ column (Higgins) using an eluent gradient (A = 2% 5 mM ammonium acetate, B = water, C = acetonitrile) consisting of an initial 2 min period of 5% A, 90% B, 5% C, followed by a linear increase to 5% A, 0% B, 95% C at 6 min and held for the remaining 4 min. For both methods, the mobile phase flow rate was held at 0.35 mL/min for the first 1.5 min, increased to 0.5 mL/min and held for the final 4 min. The high-resolution mass spectrometer was operated at 120,000 resolution and mass-to-charge ratio (*m*/*z*) range 85–1275. Probe temperature, capillary temperature, sweep gas and S-Lens RF levels were maintained at 200 °C, 300 °C, 1 arbitrary unit (AU), and 45, respectively, for both polarities. Additional source tune settings were optimized for sensitivity using a standard mixture, positive tune settings for sheath gas, auxiliary gas, sweep gas and spray voltage setting were 45 AU, 25 AU and 3.5 kV, respectively; negative settings were 30 AU, 5 AU and −3.0 kV. Maximum C-trap injection times of 100 milliseconds and automatic gain control target of 1 × 10^6^ for both polarities. During untargeted data acquisition, no exclusion or inclusion masses were selected, and data were acquired in MS1 mode only. The chromatogram (Appendix A) shows the metabolites separation in positive and negative ionization modes.

### 2.4. Data Preprocessing and Analysis

The lipidomics raw data were processed using MZmine 2.31 [33,34]. All mass spectral data with a signal-to-noise ratio > 10 and mass intensity > 1.0E4 were further processed according to the following steps. Briefly, processing includes mass detection and chromatogram building, chromatogram deconvolution, deisotoping, alignment, gap filling, and filtering. The final preprocessing of raw data contained information on mass, retention time, lipid species, and intensity. The processed features were annotated using exact mass information within ±5 ppm mass accuracy using the LIPID MAPS online server [35]. Further, lipid species were confirmed by the utilizing DDA data in LipidMatch 2017_5_3 version [36] and LipidSearch 4.1.28 (Thermo Fisher Scientific Inc., San Jose, CA, USA). Finally, the identified lipid species along with exact mass-based annotated features and unknown features were statistically analyzed to investigate the change in lipid profiles among different donor samples due to anticoagulant use and/or collection conditions. All the lipids identified based on exact mass are represented in the format “LIPIDMAPS_number” and the unknowns are labeled as “Unknown_number” for brevity. The LIPID MAPS nomenclature system [37] is used to name the lipids reported in this study and identified lipids are abbreviated as reported by the identification software [36]. For lipidomics, a total of 3,881 features in positive mode and 803 features in negative mode were extracted after preprocessing the raw mass spectral data. The identified features covered diverse lipid classes, such as lysophospholipids, phospholipids, cholesteryl esters (CEs), triacylglycerols (TGs), diacylglycerols (DGs), sphingolipids, and acylcarnitines (AcCas). The lipidomics data were quantile normalized, generalized log transformed and Pareto scaled whereas the metabolomics data were quantile normalized, log2 transformed and auto-scaled before performing downstream statistical analyses.

The metabolomics raw data were converted to open format (netCDF) using Xcalibur software (ThermoScientific, Waltham, MA). Data were extracted using apLCMS [38] as *m*/*z* features and batch-corrected with xMSanalyzer [39], where an *m*/*z* feature is defined as an accurate mass *m*/*z* with associated retention time (RT; in seconds) and intensity. Quantification results for each of the samples and external controls were recorded as LC-MS peak area. The data was tentatively annotated by xMSannotator [40]. The lipidomics and metabolomics preprocessed data were normalized and statistically treated on Metaboanalyst 4.0 [41,42]. From the metabolomics data, a total 7635 and 8519 features were extracted from positive and negative ionization modes, respectively, among which 178 positive mode and 126 negative mode features were annotated based on exact mass within ±5 ppm mass accuracy. All identified and non-identified features were considered for further analysis using univariate and multivariate approaches to study the metabolic profile difference among different time points and anti-coagulants used. The identified lipid data in positive and negative ionization modes were normalized and statistically analyzed to infer the differences in lipid profiles between the sample groups. The metabolomics and lipidomics data were analyzed by using one-way analysis of variance (ANOVA) followed by post-hoc analysis using Fisher’s Least Significant Difference (LSD) test. Principal component analysis (PCA) was performed to study sample clustering and outliers.

## 3. Results

Seventy plasma samples from five different healthy donors were grouped and sub-grouped based on anticoagulant type and storage time for statistical analysis of the preprocessed mass spectral features.

A detailed scheme of the analytical strategies is presented in Figure 1. The statistical analyses were performed by two different strategies—donor-matched and donor-unmatched for positive and negative mode data—and a similar analytical strategy was followed for the metabolomics data. In the donor-matched strategy, each donor was analyzed separately by grouping storage time and anticoagulant used. Thus, statistical treatment was undertaken for five donors separately to determine the effect of storage time and anticoagulants in the plasma lipid profile. The donor-unmatched samples were analyzed to study the effect of storage time and anticoagulants collectively, including all the donors under study.

Principal component analysis (PCA) was performed for the donor-unmatched samples by grouping the samples according to storage time and the anticoagulants used. Samples did not group based on storage time in the PCA, indicating minimal or no contribution of storage time related variance on sample clustering. Rather, the variance between donors contributed far greater to the PCA than storage time (Figure 2A). The One-way Analysis of Variance (ANOVA) with Fisher’s LSD test corroborated the PCA observations that no lipid species were significantly different across different storage times using a false discovery rate (FDR) of 0.05 except for one unknown feature (Figure 2C). Similar results were obtained for negative mode data. The PCA analysis of donor-unmatched samples for positive mode data based on anticoagulant grouping did not show clustering based on anticoagulant rather the clustering was dominantly based on donors’ lipid profile (Figure 2B). Nonetheless, the One-way ANOVA analysis with post-hoc Fisher’s LSD test showed 347 features significantly different between anticoagulants (Figure 2D, Appendix A), among which 22 were MS/MS confirmed lipids, 138 were exact mass-based annotated lipids, and 187 were unknown features. Similarly, the ANOVA analysis with Fisher’s LSD test of negative mode analysis showed 78 features to be significantly different at an FDR of 0.05. Among these six features were MS/MS confirmed lipids, 20 were exact mass-based lipid annotations and 52 were unidentified features (refer to Appendix A). Of the MS/MS confirmed positive mode lipids, most were glycerophospholipids with phosphatidylethanolamines (PEs) as the dominant class. The other confirmed lipids belonged to cholesteryl esters (CEs), acylcarnitine (AcCa), triacylglycerols (TGs), and coenzyme Q9. All MS/MS confirmed lipids were found to be higher in EDTA tubes except AcCa(18:2) [M + H] and oxTG(16:0_18:1_18:3(OO)) [M + NH_4_] which were found to be higher in CPT tubes. Of the 6 MS/MS confirmed negative mode lipids, three were PEs, two were sphingomyelins (SMs), and one was phosphatidylcholine (PC).

Further analyses were performed with donor-matched groups to study the effect of anticoagulant and the time of storage. In a donor-matched strategy using positive mode data, PCA grouped samples based on anticoagulant, CPT vs. EDTA/EDTA_4C, indicating anticoagulant is driving differential lipid profiles (Figure 3). Further investigation of differentially abundant lipids was performed by one-way ANOVA in donor-matched samples grouped by anticoagulants used. Among five donor-matched groups, all groups showed a significant effect of anticoagulants (Figure 4, Appendix A) in both ionization modes, with the exception of D14. All significant MS/MS confirmed positive and negative mode lipids in D11 were found to be higher in EDTA tubes with PE being the dominant lipid class. Other lipids include DG(18:0_18:1) [M + NH_4_] and PS(18:0_20:4) [M + H] in the positive mode. Similar results were observed in the D12 positive mode data. All significant confirmed lipids were higher in EDTA tubes and all of them were PEs except PS(18:0_20:4) [M + H]. The positive mode D13 data showed the maximum number of confirmed lipids that were significantly different between tube types including phospholipids, AcCa, sphingolipids, and TGs. Again, PE was the most differentially abundant lipid class. All the lipids were higher in EDTA containing tubes except PE(36:0) [M + H], OxTG(16:0_18:1_18:3(OO)) [M + NH4], PG(17:0_17:0) [M + H]. To note, PG (17:0_17:0) [M + H] was one of the internal standards spiked into the samples. All the differentially abundant lipids in the D13 negative mode data were phospholipids (PC and PE) and were more abundant in EDTA containing tubes (Appendix A). Donor D15 had all the significant confirmed lipids higher in EDTA containing tubes except OxTG(16:0_18:1_18:3(OO))[M + NH_4_] which was more abundant in the CPT tube. The differentially abundant lipids in D15 EDTA tubes include phospholipids (PE, PI, and PS), CE, and ceramides. No confirmed lipid was found to be differentially abundant in D15 negative mode data though 28 unknown features were found to be significantly different between anticoagulants (Appendix A). The ANOVA analysis with a Fisher LSD test of donor-matched samples grouped by storage time did not show any difference in lipid profile for positive mode data except for donor D12 where 5 lipids showed significant difference at FDR 0.05 out of which 3 lipids were identified only by exact mass based. Similar analysis with negative mode lipidomics data showed a significant effect of storage time on donors D11 and D12 with 110 (6 MS/MS confirmed) and 262 (36 MS/MS confirmed) differentially abundant lipids, respectively (FDR 0.05). Donors D13 and D14 did not show a significant effect whereas D15 showed one lipid, PE(16:0_22:6) [M − H] significantly different at FDR 0.05 (Appendix A). The effect of storage time on lipids was prominent in negative mode data when analyzed by the donor-matched strategy. Figure 5 shows the overall lipid profile difference within donors in negative ionization mode grouped by storage time (Figure 5).

Similar analyses were performed for positive and negative ionization mode metabolomics data. All the statistical analyses were performed by using Metaboanalyst 4.0 [43]. Missing imputation was performed by calculating half of the minimum positive values in the original data. The data were quantile normalized, log2 transformed, and auto-scaled. A principal component analysis was performed followed by ANOVA to detect differentially abundant metabolites due to the effect of storage time and anticoagulants on plasma samples.

For the donor-unmatched strategy, the PCA on time-grouped data showed two clusters of samples based on the anticoagulant contrary to storage time (Figure 6A). Further, the ANOVA with post hoc Fisher LSD test showed three significantly different unknown metabolites for samples stored at different time points in positive mode data (Figure 6C). For the donor-unmatched storage-time grouped negative mode data, no significantly different lipids were observed. Similarly, the anticoagulant-grouped samples showed PCA clustering based on anticoagulant type (Figure 6B) and the ANOVA with post hoc Fisher LSD analyses showed 2867 (Figure 6D, Appendix A) and 2341 features (Appendix A) to be differentially abundant in positive and negative mode data, respectively. Differentially abundant known metabolites (at FDR 0.05; 52 known metabolites in positive and 21 known metabolites in negative mode) spanned multiple biochemical pathways. Most of the differentially abundant metabolites were higher in EDTA-containing plasma samples whereas fewer metabolites were enriched in citrate-containing plasma samples.

The donor-matched analysis was performed by grouping samples according to storage time and anticoagulants. For each donor, PCA and ANOVA with Fisher LSD analyses were performed to observe the effect of storage time and anticoagulants. PCA showed clustering of samples based on anticoagulants rather than times of storage (Figure 7). Very few metabolites were differently abundant between different storage times at FDR 0.05. The donors D11, D12, D13, D14, and D15 showed four, one, zero, six, and one unknown features in positive mode and four, 55, seven, five, and nine unknown features in negative mode to be differentially abundant, respectively, at FDR 0.05 (Appendix A). Figure 8 shows the heatmap for donor-matched storage time analysis result for negative mode metabolomics data. One way ANOVA with post hoc analysis at FDR 0.05 showed an array of metabolites to be differentially abundant between CPT and EDTA-containing samples (Figure 9). The donors D11 and D12 showed 1518 (31 identified) and 1055 (19 identified) features in positive (Appendix A) and 1390 (11 identified) and 1033 (eight identified) (Appendix A) features in negative modes respectively, different between CPT- and EDTA-containing tubes. Similarly, the donors D13, D14 and D15 showed 1463 (22 identified), 1917 (31 identified) and 1403 (27 identified) metabolic features significantly different between anticoagulants at FDR < 0.05 (refer to the Figure 9 and Appendix A for the significantly different features observed in the metabolomics data). The negative mode metabolomics data showed 1361 (14 identified), 1746 (11 identified) and 1344 (10 identified) significantly different features in D13, D14 and D15 (Appendix A) donors at FDR < 0.05. Taken together, the lipid profile is affected predominantly by an anticoagulant type followed by minor contributions from storage time. It has also been observed that the storage time affects the lipid and metabolites profile significantly in donor-matched samples however the effect is nullified in the donor-unmatched analysis that might potentially explain the contribution of donor’s lipid or metabolite profiles in variation. Donor lipid profiles also contribute to the variance as observed in PCA analysis (Figure 2A,B) whereas the anticoagulant effect is the main determinant of variance in the metabolomics data (Figure 5A,B).

## 4. Discussion

Different physical and chemical conditions are encountered during clinical sample collection, handling, processing and storage which may affect lipid and metabolite composition and abundance in a manner unrelated to the physiology or pathophysiology under study. These sample handling perturbations have the potential to confound attempts to identify biologically meaningful observations; therefore, the source and magnitude of these non-biological effects must be identified prior launching a mass spectrometry based ‘omics clinical study. Table 1 summarizes the findings from the previous studies on the effect of different physical and chemical conditions in various biological samples. This study evaluated sample handling bias on the metabolome and lipidome of healthy donors prior to launching VTEU vaccine studies conducted in locations that may generate inconsistent sample handling. This is a common issue since sample processing and analysis are usually performed at locations geographically distinct from collection sites. We evaluated the effect of different anticoagulants and storage conditions on plasma lipids and metabolites plasma samples to make recommendations for the sample collection protocol.

The statistical analysis was performed in total unmatched plasma samples and then further analysis was performed by matching the donors and grouping the samples based on the tube type, sample handling time and temperature. Both the donor-matched and donor-unmatched analysis showed significant differences in lipid profiles between CPT and EDTA-containing samples. Phospholipids were the most affected lipid class. Other affected lipid classes included AcCa, sphingolipids, DGs, TGs, coenzyme, and CEs. Similar studies have been conducted to study the effect of anticoagulants in different biological specimens and mixed results have been reported regarding their effects on metabolite and lipids profiles. For example, PCs and PEs, SMs, TGs and cholesterol were found in higher levels in EDTA-containing plasma vs. citrate-contains plasma which is consistent with our findings [44]. Cholesterol was not covered in our study; however, cholesteryl esters were found to be highly abundant in EDTA-containing plasma relative to citrate-containing plasma. Similarly, Gonzalez-Covarrubias et al. (2013) showed lysophosphatidylcholine (LPC), PCs, SMs, many PEs, and TGs were found to be higher in EDTA-containing plasma samples compared to citrate or heparin suggesting differences may be driven by anticoagulant chemistry, plasma pH, extraction or the electrospray ionization efficiencies of lipids. [53]. The anticoagulant concentration may cause sample dilution due to osmotic effects [54,55,56,57] that can potentially affect the metabolite abundance in LC-MS analysis of plasma samples. The osmotic effect draws water out of the red blood cells immediately after contact with blood resulting in dilution of plasma lipids below their actual concentration. This effect, however, was not influenced by processing temperature [56,57].

Heparin is an anti-thrombin activator and EDTA and citrate are metal ions chelators. These chemical components can inactivate the fibrinogen-initiated clotting cascade forming a clot-free blood component, plasma [58]. Though heparin is not detected in the untargeted metabolomics assays, the lithium heparin plasma can form unwanted lithium adducts [59] and boost the ionization efficiency of phospholipids and triacylglycerols including plastic polymers in the collection tube potentially causing serious matrix effects [21]. Several amino acids including aspartate, histidine, and glutamine were observed at higher levels in EDTA-containing plasma and serine was observed in higher levels in citrate-containing plasma in our study. This is in line with previous studies which showed higher amino acid and amine levels in EDTA plasma [44]. The concentrations of phosphoethanolamine, citrulline, cysteine, and tryptophan were observed to increase in EDTA-containing plasma and decrease in citrate-containing plasma when compared with heparin-treated control plasma [27]. As expected, citrate was observed to be higher in citrate-containing tubes. Signal response in LC/MS can also be affected by plasma components, LC tubing interactions, or column active site blocking, etc. Similarly, technical issues might be encountered with small molecule anticoagulants such as a reduction in column lifetime and enhancement of ion suppression, thus making heparin the preferable anticoagulant for metabolomics [47].

In our study, plasma samples were collected and stored in EDTA-containing tubes for different time points at 4 °C and room temperature while citrate-preserved samples were stored only at room temperature. The storage times between sample collection and processing were 0 h, 2 h, 4 h, 8 h and 24 h (note: EDTA-containing samples at room temperature and 4 °C have baseline sample as 0-h samples). One-way ANOVA and Fisher LSD analyses were performed by grouping and ungrouping the plasma samples by donors. Our analyses showed no significant change in lipid profile across storage times in overall donor-matched analysis however the donor-matched analysis showed several different class of lipids to be affected by storage time. The lipid classes like PC, SM, LPE, and PE were found to be affected due to storage time where the inverse relation between lipid abundance and storage time was observed. Nonetheless, many of the plasma and lipids metabolites were affected by storage temperature. Most of the significantly affected lipids were phospholipids (PEs and LPCs) and were highly abundant in EDTA-treated plasma stored at room temperature. Very few lipids were shown to be higher at 4 °C and none were MS/MS confirmed. Similar results were observed in metabolites levels where EDTA-treated plasma stored at room temperature show many significantly different metabolites, while very few known metabolites are differentially abundant in EDTA-containing plasma at 4 °C. There is sparse information about the effect of temperature and storage conditions at the metabolite or lipid level though a few studies have been performed to study the effect of long-term storage and temperature on total lipid content in plasma [45,60,61]. However, studies showed inconsistent results on the effect of storage and temperature on plasma and serum lipid content. Serum sample storage at −80 °C over a year led to a decrease in total cholesterol and triglycerides [45,60,61,62]. The triglyceride contents of very low-density lipoprotein (VLDL) was found to be stable for 11 weeks of storage at −20 °C but decreased beyond this time window [63]. Mean high-density lipoprotein (HDL)-cholesterol levels remain unchanged when stored at 4–6 °C for 4 days. However, HDL-cholesterol tended to decrease by day 7 and increase by day 14 but this depended upon the initial plasma concentration [64]. In a recent study, 24% of blood metabolites were found to be significantly affected when stored for 6 h at room temperature and 21% of serum metabolites were significantly affected when stored for room temperature for 24 h [65]. The above-mentioned studies are focused on lipoproteins, cholesterol and triglycerides and lack information about the effect of storage and temperature at the lipid species level. With recent advancements in GC/MS and LC/MS, it is now possible to study the lipid content of the plasma at individual lipid species level enabling a deeper understanding the effect of various physical and chemical variables on plasma lipid and metabolite profiles. GC-MS based untargeted metabolomics studies of sputum samples maintained stability for 8 weeks at −20 °C and −80 °C while metabolite abundance changed after one day at 4 °C [23]. An NMR based metabolomics study showed that the change in energy-related metabolites was retarded when plasma samples were processed on ice [17]. In another study, EDTA-containing blood samples showed stability up to 4 h when processed in ice-cold water and withstand the effects of 4 freeze-thaw cycles with only slight changes in metabolome; however, hypoxanthine and sphingosine 1-phosphate were significantly affected with an increase of 1810% and 710% respectively when stored at room temperature for 2 h [22]. This study did not assess the effect of the freeze-thaw cycles and many commonly used anticoagulants; however, other studies [19,49] recommend the use of heparin for LC-MS-based metabolomics contrasting the recommendation made by Yuille et al. (2010) [66] which details a wide range of freeze-thaw cycle (2–10 freeze-cycles) without significant effect on metabolites [22,23,67,68,69] One LC-MS/MS-based study showed a linear increase (approximately 50%) of heparin whole blood choline concentration during storage for >4 h whereas the choline concentration in EDTA whole blood increased for an hour and then stabilized [70]. Stability assessment of total cholesterol, high-density lipoprotein cholesterol and triglycerides of human serum stored at −15 °C for 18 weeks showed no significant change in concentration [71]. Similarly, photometry based assays showed lipids and polar metabolites such as bilirubin, glucose, cholesterol, triglycerides, and high-density lipoprotein remained stable at −20 °C for up to three months and withstood the effect of up to 10 freeze-thaw cycles [67]. Several metabolites such as betain, sarcosine, and creatinine were relatively stable whereas methionine and most B vitamins were susceptible to oxidative degradation during long term storage at −25 °C [21]. Long term storage of unsaturated fatty acids, especially eicosapentanoic acid (EPA) and docosahexanoic acid (DHA), increases oxidative damage and can be reduced by freezing and treating blood with heparin due to its antioxidant properties [21,72,73]. Most hormones should be collected in EDTA-containing tubes and transported and stored at 4 °C with the exception of adrenocorticotrophic hormone (ACTH) which requires frozen transport and storage [26]. NMR based metabolomics showed that urine and serum can be stored up to 24 h at 4 °C with undetectable changes in metabolite profile [49].

Anticoagulants affect the metabolome and lipidome of biological specimens; however, findings from different studies are not consistently aligned offering no concrete general guidelines for sample collection and processing prior to LC-MS (Summarized in Table 1). Studying collection-related variables upon the lipidome and metabolome provides insights into understanding the potential for sample collection and processing artifact. This can help design collection and processing techniques to minimize these potential disturbances in clinical studies (e.g., vaccine studies). Based on our findings the following recommendations were made for sample collection, processing, handling, and preservation. We recommend using EDTA as an anticoagulant over citrate. It is always recommended to limit the freeze-thaw cycle of biological samples. We recommend low-temperature processing and handling for blood though no significant temperature-induced changes were observed in this study. We did not observe serious effects on lipidome and metabolome profile until 24 h prior to flash freezing. However, it is highly recommended that the blood collection and processing should be completed performed at the lowest possible temperature and preserved at −80 °C within 24 h. It is important that metabolomics and lipidomics studies report the specimen collection parameters such as tube type, processing time and temperature to ensure that results can be adjusted to account for artificial collection or processing-induced variability.

## 5. Conclusions

The effect of storage temperature, storage time, and anticoagulant on the plasma lipidome and metabolome were studied in healthy donors. From this study, we found that the lipidome was not significantly affected when plasma was stored at 4 °C or room temperature until 24 h. However, anticoagulant had significant effects in plasma lipid profiles. Very few metabolites were affected due to temperature differences. Several studies have shown inconsistent findings in terms of the effect of temperature, storage time, anticoagulants, and freeze-thaw cycles on metabolites and lipids profiles though many of the studies consistently align with the conclusion of their findings. Nonetheless, the effect of various physical and chemical parameters on biological specimens is debatable and without solid explanation. It is possible that several changes occur during sample collection, processing, and handling that can potentially introduce unwanted variability. Many of those critical parameters depend upon the distance of the sample collection to sample processing sites, availability of resources for sample processing and preservation, time lag during sample collection, processing, shipping, and extraction. Thus the efforts should be directed to find study-dependent critical parameters. Depending on the sample type, analytical methods, and overall processing steps, it is the researchers’ responsibility to determine critical parameters that could introduce variability and need to be investigated before initiating the clinical study. Overall, we recommend that the samples should be collected in EDTA-containing tubes and that all the processing work before freezing at −80 °C should be completed within 24 h while maintaining the temperature of 4 °C. This study did not investigate the mechanism behind the effects of anticoagulant type on plasma lipids though several potential explanations have been proposed. Further studies should be directed toward understanding the mechanism behind the differential anticoagulant effects, temperature effects, and storage effects on plasma lipids and metabolites.

## Figures and Tables

**Figure 1 biomolecules-09-00200-f001:**
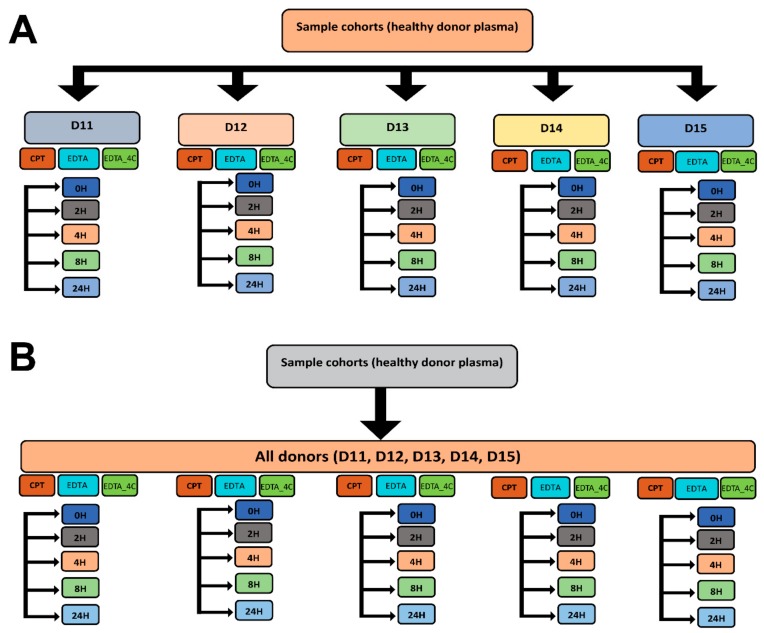
Statistical analysis strategies for untargeted lipidomics. (**A**) donor-matched strategy and (**B**) donor-unmatched strategy. The study comprises five donors (D11, D12, D13, D14, and D15), two different anticoagulants sodium citrate and potassium ethylenediaminetetraacetic acid represented as CPT and EDTA respectively, two different temperatures (4 °C and room temperature) and five different storage times before processing (0H, 2H, 4H, 8H, and 24H). The 0H EDTA samples represent the baseline for both EDTA containing samples at 4 °C and room temperature.

**Figure 2 biomolecules-09-00200-f002:**
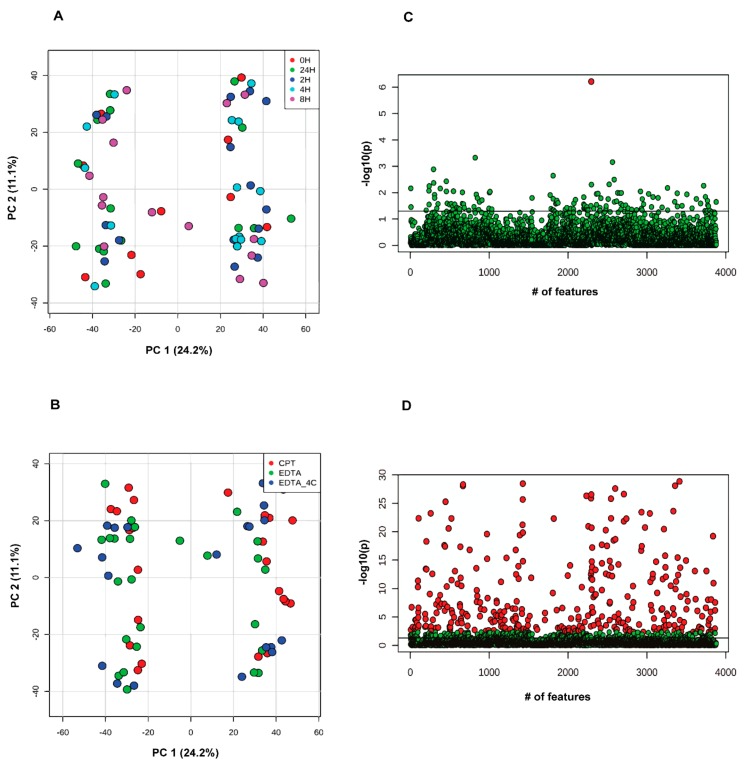
Principal component analysis in donor-unmatched samples from lipidomics data acquired in positive ionization mode. Samples are grouped by (**A**) storage time and (**B**) anticoagulants used. Analysis of variance with Fisher’s Least Significant Difference (LSD) test at adjusted p-value at 0.05 grouped by (**C**) storage time (red dot show *p*-value < 0.05) and (**D**) anticoagulants (red dots show *p*-value <0.05). Note: The *Y*-axis scale of scatter plot is automatically adjusted by Metaboanalyst based on the range of −log10(p). The line on the scatter plot demarcates the −log10(0.05) value on *Y*-axis.

**Figure 3 biomolecules-09-00200-f003:**
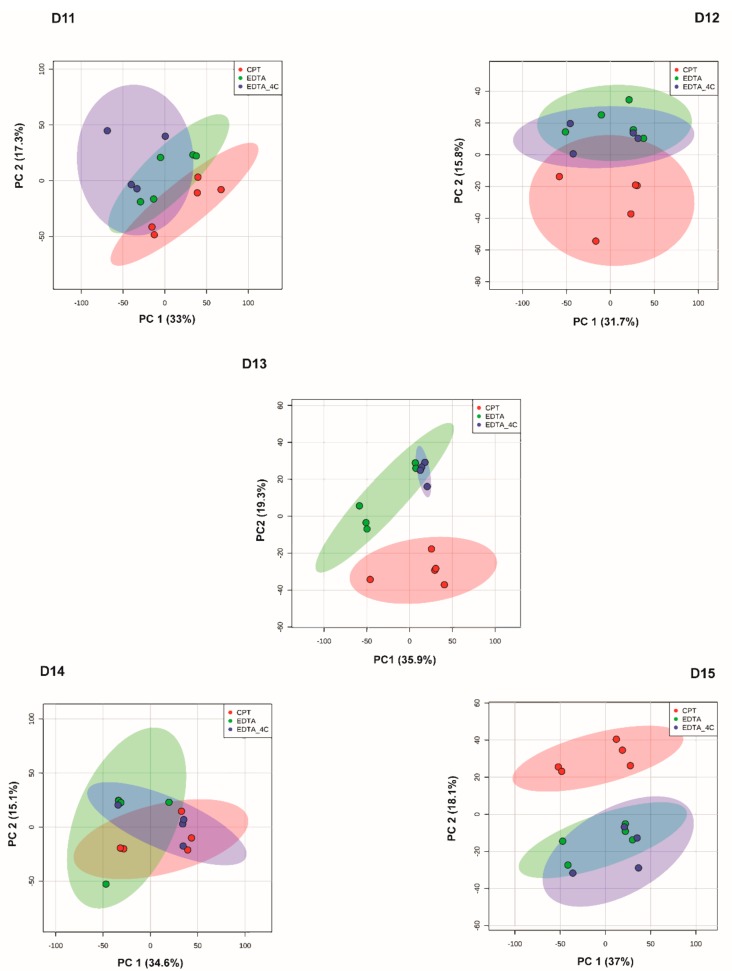
Principal component analysis among donor-matched samples grouped by anticoagulants used for positive mode lipidomics data. Principal component analysis (PCA) showed differences in lipids between CPT and EDTA preservatives. The colored ellipses represent the 95% confidence interval.

**Figure 4 biomolecules-09-00200-f004:**
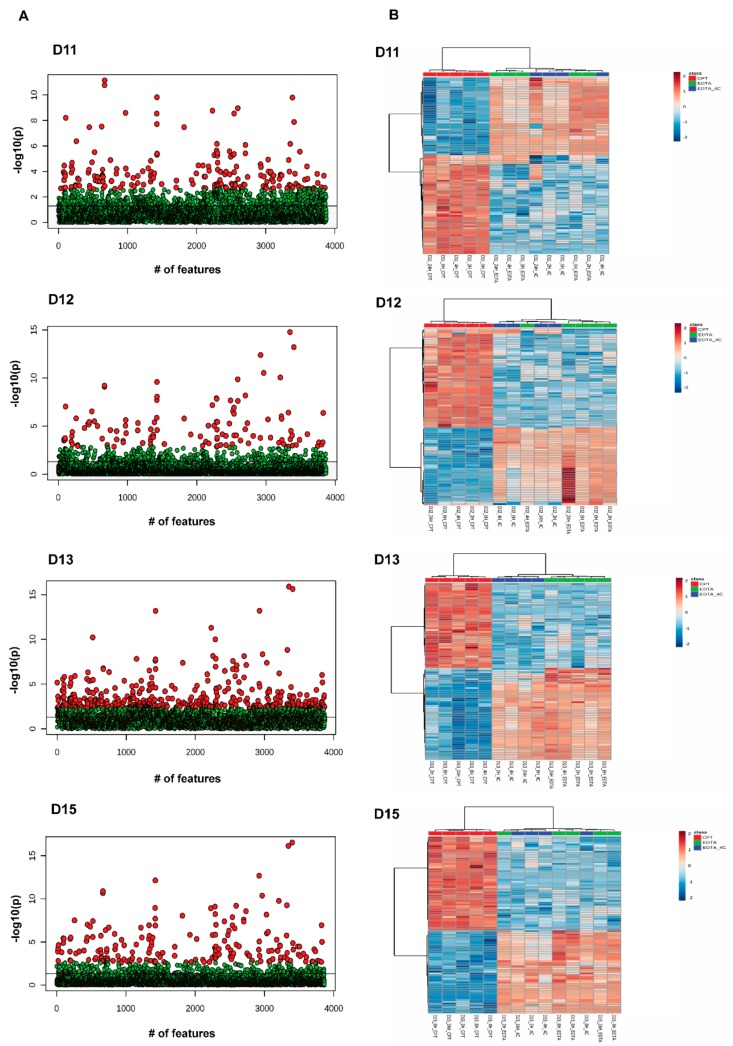
One-way ANOVA analysis of donor-matched lipidomics data. (**A**) Scatterplot showing One-way analysis of variance among donor-matched samples grouped by anticoagulants (red dots show *p*-value < 0.05). (**B**) Two-way hierarchical clustering analysis using Euclidean distance and ward.D clustering algorithm among donor-matched samples grouped by anticoagulants. The heatmap shows the top 100 lipid species that are different between CPT and EDTA containing tubes. Note: The *Y*-axis scale of scatter plot is automatically adjusted by Metaboanalyst based on the range of −log10(p). The line on the scatter plot demarcates the −log10(0.05) value on *Y*-axis.

**Figure 5 biomolecules-09-00200-f005:**
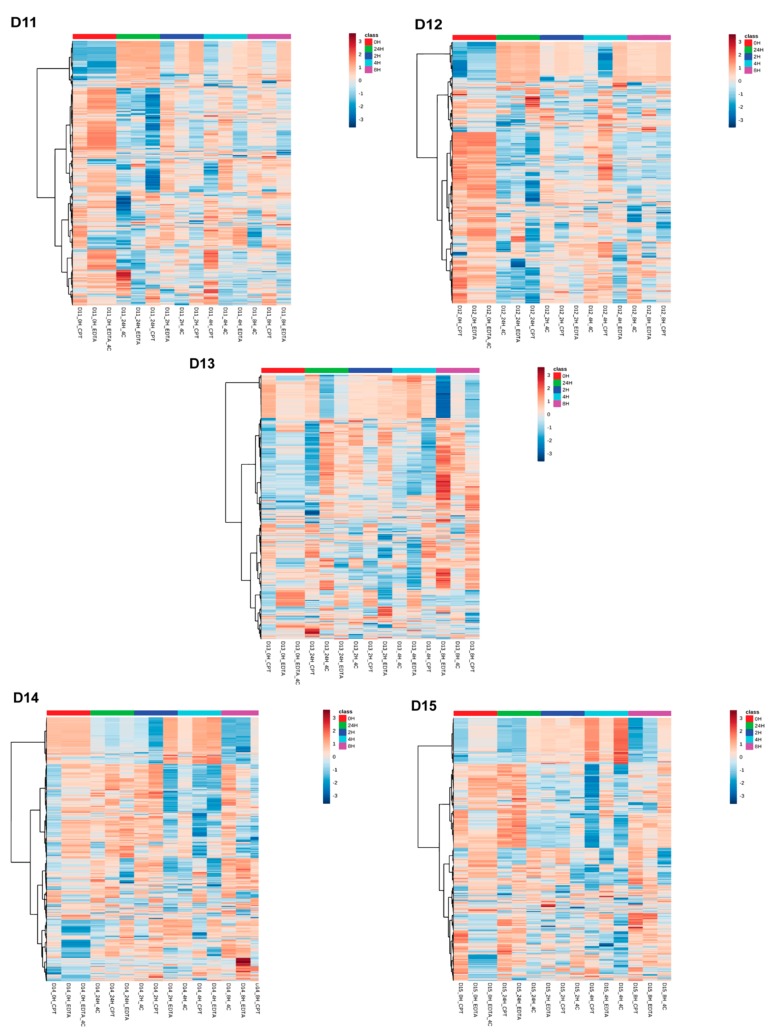
Two-way hierarchical clustering analysis using Euclidean distance and ward.D clustering algorithm among donor-matched samples grouped by storage time. The heatmap shows overall lipid profile differences in negative ionization mode grouped by storage time in donor-matched samples.

**Figure 6 biomolecules-09-00200-f006:**
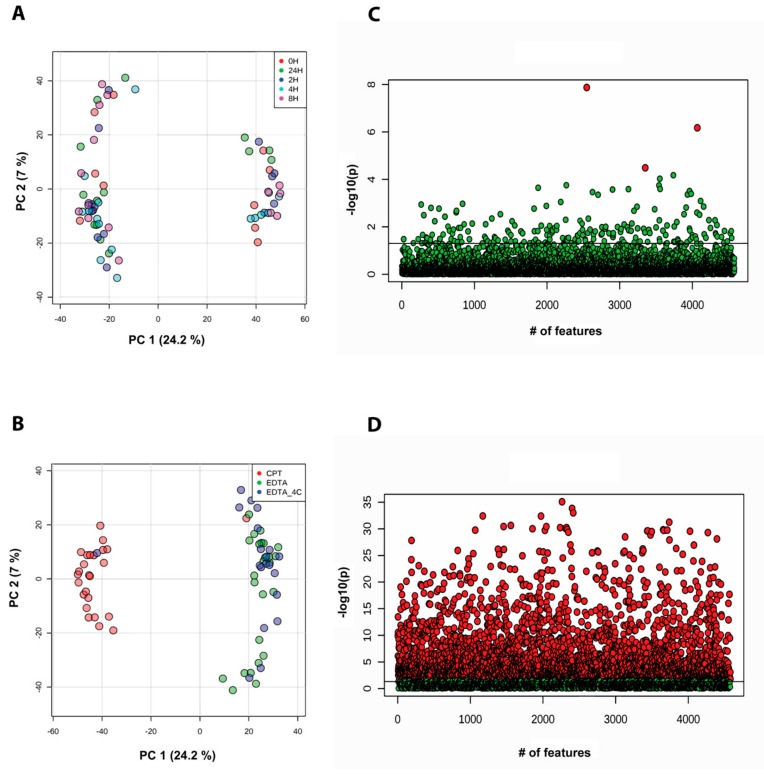
Principal component analysis in donor-unmatched samples from metabolomics data acquired in positive ionization mode. Samples are grouped by (**A**) storage time and (**B**) anticoagulants used. Analysis of variance with Fisher’s LSD test at adjusted p-value at 0.05 grouped by (**C**) storage time and (**D**) anticoagulants (red dots show *p*-value < 0.05). Note: The *Y*-axis scale of scatter plot is automatically adjusted by Metaboanalyst based on the range of −log10(p). The line on the scatter plot demarcates the −log10(0.05) value on *Y*-axis.

**Figure 7 biomolecules-09-00200-f007:**
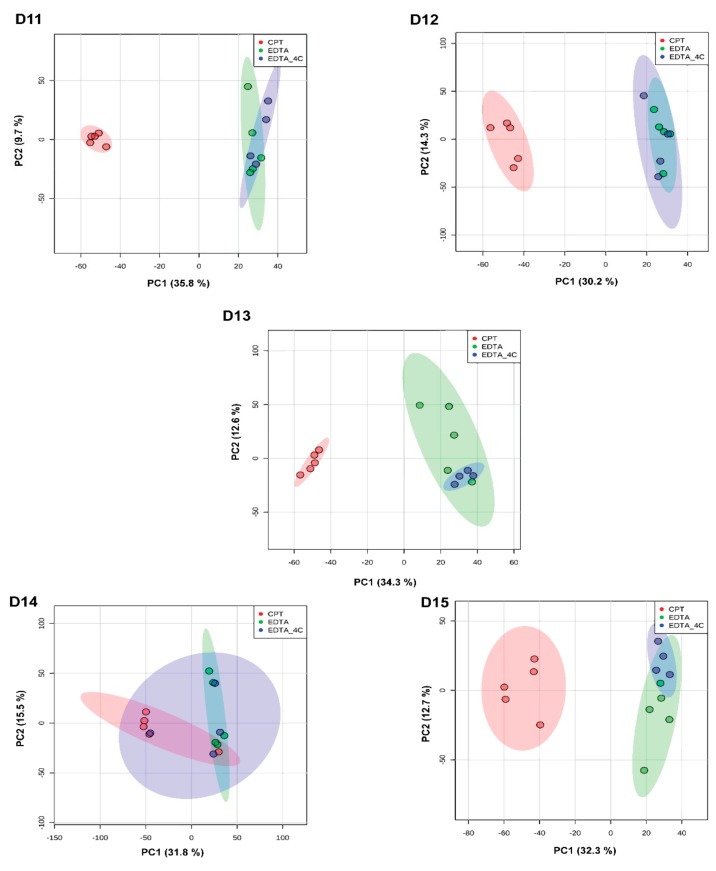
Principal component analysis among donor-matched samples grouped by anticoagulants for metabolomics data acquired in positive ionization mode. The PCA analyses showed differences in lipid profiles between CPT and EDTA. The colored ellipse represents the 95% confidence interval.

**Figure 8 biomolecules-09-00200-f008:**
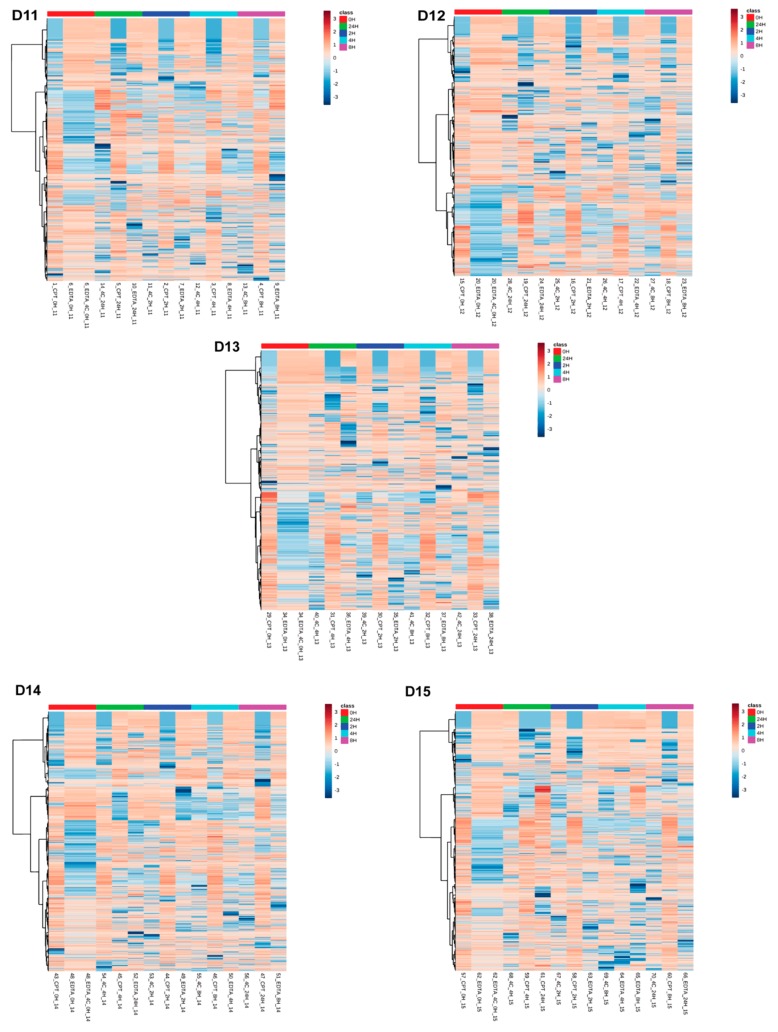
Two-way hierarchical clustering analysis using Euclidean distance and ward.D clustering algorithm among donor-matched samples grouped by storage time. The heatmap shows overall metabolite profile differences in negative ionization mode grouped by storage time in donor-matched samples.

**Figure 9 biomolecules-09-00200-f009:**
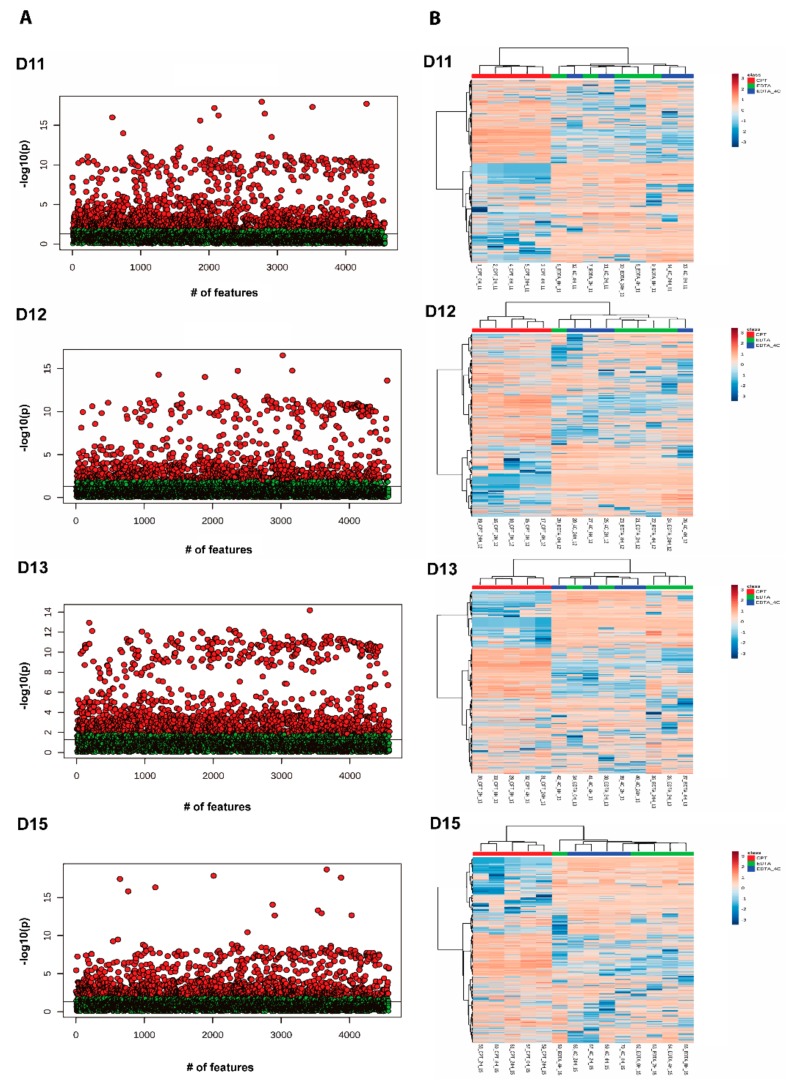
One-way ANOVA analysis of donor-matched metabolomics data. (**A**) Scatterplot showing One-way analysis of variance among donor-matched samples grouped by anticoagulants (red dots show metabolites with *p*-value < 0.05), (**B**) Two-way hierarchical clustering analysis using Euclidean distance and ward.D clustering algorithm among donor-matched samples grouped by anticoagulants. The heatmap shows overall features from metabolomics data in positive ionization mode that are different between CPT and EDTA containing tubes. Note: The *Y*-axis scale of scatter plot is automatically adjusted by Metaboanalyst based on the range of −log10(p). The line on the scatter plot demarcates the −log10(0.05) value on *Y*-axis.

**Table 1 biomolecules-09-00200-t001:** Effect of different physical and chemical parameters on lipids and metabolites in various biological samples.

Reference	Year	Material	Method	Time	Temperature	Anticoagulant	Freeze/Thaw	Other	Conclusion
Teahan et al. [17]	2006	Serum and plasma	NMR	2 h clot time	Room temperature and ice	Heparin	Yes		Variation due to individual. Freeze-thaw cycle should be minimal, no number of cycle recommended. Addition of heparin reduced the effect compared to non-heparinated serum or plasma samples. Serum clot contact time is the significant factor in introducing variation. Clotting on ice delayed the effect of temperature.
Bando et al. [18]	2010	Plasma and urine	GC-MS	NA	Room temperature and ice	Plasma: K2EDTA, Sodium heparin			Citrate, 2-oxoglutarate, hippurate, threitol, threonate elevated in 4 h pooled sample. Heparinated plasma overlapped with other endogenous metabolites like sugars potentially causing inter-sample variation. EDTA did not overlap with other endogenous metabolites. Recommended the use of EDTA plasma for GC-MS analysis.
Barri et al. [19]	2013	Serum and plasma	LC-MS	NA	NA	K2EDTA, Li-Heparin, Na-citrate			Coagulation effect on serum led to release of peptides, hypoxanthine, and xanthine and can be nullified with robust data processing. Plasma anticoagulant can lead to ion suppression or enhancement on metabolites. Anticoagulant cation can make metabolites dominant in positive ESI/MS mode. Heparin preferred over others due to no observable matrix effects. EDTA and citrate plasma elicit sodium and potassium formate cluster causing ion suppression or enhancement of the coeluting metabolites. Blood serum is an alternative to avoid the anticoagulant matrix effect in the plasma sample.
Yin et al. [22]	2013	Serum and plasma	LC-MS	2, 4, 8, and 24 h	Room temperature and ice	Heparin, EDTA	Yes	Hemolysis	L-carnitine significantly decreased after two to four cycle. No significant effect of up to two freeze-thaw cycles. Hypoxanthine, sphingosine-1-phosphate, and linolenyl carnitine significantly altered in the range of 0 h to 24 h. No change in metabolome up to 4 h when stored in ice. Sphingosine-1-P and hypoxanthine showed significant change after 4 h at room temperature. Significant hemolysis effect was observed in two species of lysophosphatidylcholines. Chemical noise pattern was observed in lithium heparinate and serum blood collection tubes. Polyethylene glycol ion cluster potentially leached out from the plastic bead in the collection tube.
Wandro et al. [23]	2017	Sputum	GC-MS		4 °C and −20 °C	NA	Yes		Aspartic acid, glycine, isoleucine, serine, and uracil abundance increased after a day when stored at 4 °C. No effect at −20 °C. No effect of one to two freeze-thaw cycle.
Haid et al. [25]	2018	Plasma	LC-MS		Long term storage at −80 °C	NA			Increase in concentration for amino acids, hexoses, butyrylcarnitine, phospholipids containing more than 40 carbon. The decrease in concentration of acylcarnitines, lysophosphatidylcholines, diacyl-phosphatidylcholines, acyl-alkyl phosphatidylcholines, and sphingomyelin.
Jorgenrud et al. [44]	2015	Plasma and serum	GC-MS		Room temperature and 4 °C	EDTA, citrate			Amino acids higher in EDTA plasma, Amines abundance higher in serum and lowest in citrate plasma. Phenolic compounds abundance highest in EDTA and lowest in citrate plasma. Total carboxylic acid higher in citrate plasma compared to EDTA, Sterols, lactic acid, and serine abundances were lower in citrate compared to EDTA. No effect of temperature on lipids.
Mei et al. [20]	2003	Serum and plasma	LC-MS			Li-Heparin, Na-Heparin, Na_2_EDTA			Li-Heparin and polymers from the container showed matrix effect.
Zivkovic et al. [45]	2009	Serum	GC-FID		4 °C, −20 °C, −80 °C				0–4% of metabolites affected in most lipid classes when stored for a week at 4 °C, −20 °C and −80 °C
Yu et al. [46]	2011	Serum and plasma	FIA-MS						Reproducibility comparatively better in plasma. Arginine, PC (38:1), LPC (16:0, 17:0, 18:0, 18:1), serine, phenylalanine, glycine were 20%–26% higher in serum compared to plasma.
Hebels et al. [47]	2013	Plasma	LC-MS	0 h to 24 h	Room temperature and −80 °C	Heparin, EDTA, citrate			No effect of storage time on metabolites. <1% metabolites significantly different at FDR<0.05.
Barton et al. [48]	2008	Serum and urine	NMR	0 h to 36 h					Plasma and urine metabolic profiles are not affected when stored at 4 °C up to 24 h.
Dunn et al. [49]	2008	Serum and urine	GC-MS	0 h to 24 h					No significant changes in metabolome was observed at two different storage time at 4 °C.
Heiskanen et al. [50]	2013	Plasma	Shotgun MS	Plasma at −80 °C monitored for 42 months				Plasma sample volume (5 and 10 µL)	The higher plasma volume provided more stability to lipid concentration. The storage time did not have an effect on lipid stability.
Gika et al. [51]	2007	Urine	LC-MS	1 month	Two temperatures −20 °C and -80 °C		Yes	Urine extract in autosampler for 20 h at 4 °C	No detectable effect on metabolites at two different temperatures for a month. Sample stable for at least 20 h at 4 °C in the autosampler. The sample can withstand one to nine freeze-thaw cycles without significant effect on metabolites.
Deprez et al. [52]	2002	Plasma	NMR	0–9 month	4 °C and room temperature				No change in metabolite profile when snap frozen and stored at -80 °C for 9 months. A significant increase was observed in tyrosine, phenylalanine and glycerol for the sample at room temperature. Choline, 3-hydroxybtyurate, acetate, glycerol slightly increase when stored at 4 °C for 3–4 days likely due to enzymatic action.

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
