# Peer review of "The Effect of Anticoagulants, Temperature, and Time on the Human Plasma Metabolome and Lipidome from Healthy Donors as Determined by Liquid Chromatography-Mass Spectrometry"

_biomolecules, 2019, doi:10.3390/biom9050200_

Round 1

Reviewer 1 Report

This manuscript evaluated the effects of anticoagulant, temperature, and time on metabolites and lipids in healthy human plasma samples. Although this is not a novel topic, this manuscript gives a nice summary of previous studies.

I have major concerns regarding the experimental design and analysis/presentation of data.

1. Both donor-matched and -unmatched samples were collected from same donors (5 subjects) (Figure 1). What is the purpose to use donor-matched and -unmatched samples? What criteria are used to differentiate donor-matched and -unmatched samples?

2. The manuscript uses PCA to separate different anticoagulants and storage temperature. However, it does not provide direct comparison of metabolite change upon storage time, anticoagulant, and storage temperature. The effect of storage time can be assessed by the fold changes of metabolites in same subject at 2, 4, 8, 24 hours compared to 0 hour using a heat map. Lack of 0-hour samples in EDTA plasma is a significant weakness in this study. Similarly, the effect of anticoagulants and storage temperature should be evaluated by the fold changes of metabolites in the plasma from same subject.

3. Does Figure 2 present the lipidomics data for donor-unmatched samples?

4. The Figure 6 does not show the storage time data for donor-matched samples.

5. The full names of abbreviations should be given when abbreviations are first used.

6. Line 328, [[47,48] should be [47,48].

7. Line 387, “deteails” should be “details”.

Reviewer 2 Report

The authors submitted a manuscript which describes mass spectrometry based non-targeted metabolomics and lipidomics analysis of blood samples. The authors compared the metabolite and lipid profiles of blood samples processed and stored at different conditions and reported results and recommendations. 

Overall the writing is clear however, some important information and reference are missing or omitted, and graphical data presentation is needed some work in order to be accepted for publication in Metabolites. This manuscript needs revision.

Below some suggestions for authors on improvements in the current manuscript: 

Omitted reference: Kamlage B, Neuber S, Bethan B, et al. Impact of Prolonged Blood Incubation and Extended Serum Storage at Room Temperature on the Human Serum Metabolome. Metabolites. 2018;8(1):6. Published 2018 Jan 13. doi:10.3390/metabo8010006  https://www.ncbi.nlm.nih.gov/pmc/articles/PMC5875996/

Move all the scatter plots with thousands  # of features to supplemental material. Present the scatter plots with the annotated features in the manuscript. Vast majority of unidentified features obviously belongs to adducts and fragments generated with in source fragmentation.

Explain why (sentence 534, page 28)  quantification results for each of the samples and external controls were recorded as LC-MS peak intensity. It is known that chromatography peaks generated with HILIC LC-MS are not sharp and narrow compare to ones generated with RP-separations. In this particular case chromatography peak area is the most accurate read out.

Present Mass chromatograms at least for pooled samples in supplemental material.

Explain how metabolites were identified. The authors (sentence 468, page 26) decsribed using MS/MS for lipids analysis. However no such an experiment is described for metabolites identification. 

Round 2

Reviewer 1 Report

My concerns for this manuscript have been addressed by the authors. It is ready to accept for publication.